# Maximum Margin Interval Trees

**Alexandre Drouin**
Département d'informatique et de génie logiciel
Université Laval, Québec, Canada
`alexandre.drouin.8@ulaval.ca`

**Toby Dylan Hocking**
McGill Genome Center
McGill University, Montréal, Canada
`toby.hocking@r-project.org`

**François Laviolette**
Département d'informatique et de génie logiciel
Université Laval, Québec, Canada
`francois.laviolette@ift.ulaval.ca`

## Abstract

Learning a regression function using censored or interval-valued output data is an important problem in fields such as genomics and medicine. The goal is to learn a real-valued prediction function, and the training output labels indicate an interval of possible values. Whereas most existing algorithms for this task are linear models, in this paper we investigate learning nonlinear tree models. We propose to learn a tree by minimizing a margin-based discriminative objective function, and we provide a dynamic programming algorithm for computing the optimal solution in log-linear time. We show empirically that this algorithm achieves state-of-the-art speed and prediction accuracy in a benchmark of several data sets.

## 1 Introduction

In the typical *supervised regression* setting, we are given set of learning examples, each associated with a real-valued output. The goal is to learn a predictor that accurately estimates the outputs, given new examples. This fundamental problem has been extensively studied and has given rise to algorithms such as Support Vector Regression (Basak *et al.*, 2007). A similar, but far less studied, problem is that of *interval regression*, where each learning example is associated with an interval $(\underline{y}_i, \overline{y}_i)$, indicating a range of acceptable output values, and the expected predictions are real numbers.

Interval-valued outputs arise naturally in fields such as computational biology and survival analysis. In the latter setting, one is interested in predicting the time until some adverse event, such as death, occurs. The available information is often limited, giving rise to outputs that are said to be either un-censored ($-\infty < \underline{y}_i = \overline{y}_i < \infty$), left-censored ($-\infty = \underline{y}_i < \overline{y}_i < \infty$), right-censored ($-\infty < \underline{y}_i < \overline{y}_i = \infty$), or interval-censored ($-\infty < \underline{y}_i < \overline{y}_i < \infty$) (Klein and Moeschberger, 2005). For instance, right censored data occurs when all that is known is that an individual is still alive after a period of time. Another recent example is from the field of genomics, where interval regression was used to learn a penalty function for changepoint detection in DNA copy number and ChIP-seq data (Rigaill *et al.*, 2013). Despite the ubiquity of this type of problem, there are surprisingly few existing algorithms that have been designed to learn from such outputs, and most are linear models.

Decision tree algorithms have been proposed in the 1980s with the pioneering work of Breiman *et al.* (1984) and Quinlan (1986). Such algorithms rely on a simple framework, where trees are grown by recursive partitioning of leaves, each time maximizing some task-specific criterion. Advantages of these algorithms include the ability to learn non-linear models from both numerical and categorical data of various scales, and having a relatively low training time complexity. In this work, we extend the work of Breiman *et al.* (1984) to learning non-linear interval regression tree models.

## 1.1 Contributions and organization

Our first contribution is Section 3, in which we propose a new decision tree algorithm for interval regression. We propose to partition leaves using a margin-based hinge loss, which yields a sequence of convex optimization problems. Our second contribution is Section 4, in which we propose a dynamic programming algorithm that computes the optimal solution to all of these problems in log-linear time. In Section 5 we show that our algorithm achieves state-of-the-art prediction accuracy in several real and simulated data sets. In Section 6 we discuss the significance of our contributions and propose possible future research directions. An implementation is available at `https://git.io/mmit`.

## 2 Related work

The bulk of related work comes from the field of survival analysis. Linear models for censored outputs have been extensively studied under the name *accelerated failure time* (AFT) models (Wei, 1992). Recently, L1-regularized variants have been proposed to learn from high-dimensional data (Cai *et al.*, 2009; Huang *et al.*, 2005). Nonlinear models for censored data have also been studied, including decision trees (Segal, 1988; Molinaro *et al.*, 2004), Random Forests (Hothorn *et al.*, 2006) and Support Vector Machines (Pölsterl *et al.*, 2016). However, most of these algorithms are limited to the case of right-censored and un-censored data. In contrast, in the interval regression setting, the data are either left, right or interval-censored. To the best of our knowledge, the only existing nonlinear model for this setting is the recently proposed Transformation Tree of Hothorn and Zeileis (2017).

Another related method, which shares great similarity with ours, is the $L1$-regularized linear models of Rigaill *et al.* (2013). Like our proposed algorithm, their method optimizes a convex loss function with a margin hyperparameter. Nevertheless, one key limitation of their algorithm is that it is limited to modeling linear patterns, whereas our regression tree algorithm is not.

## 3 Problem

### 3.1 Learning from interval outputs

Let $S \stackrel{\text{def}}{=} \{(\mathbf{x}_1, \mathbf{y}_1), ..., (\mathbf{x}_n, \mathbf{y}_n)\} \sim D^n$ be a data set of $n$ learning examples, where $\mathbf{x}_i \in \mathbb{R}^p$ is a feature vector, $\mathbf{y}_i \stackrel{\text{def}}{=} (\underline{y_i}, \overline{y_i})$, with $\underline{y_i}, \overline{y_i} \in \overline{\mathbb{R}}$ and $\underline{y_i} < \overline{y_i}$, are the *lower* and *upper limits* of a *target interval*, and $D$ is an unknown data generating distribution. In the interval regression setting, a predicted value is only considered erroneous if it is outside of the target interval.

Formally, let $\ell : \mathbb{R} \to \mathbb{R}$ be a function and define $\phi_\ell(x) \stackrel{\text{def}}{=} \ell[(x)_+]$ as its corresponding hinge loss, where $(x)_+$ is the positive part function, i.e. $(x)_+ = x$ if $x > 0$ and $(x)_+ = 0$ otherwise. In this work, we will consider two possible hinge loss functions: the linear one, where $\ell(x) = x$, and the squared one where $\ell(x) = x^2$. Our goal is to find a function $h : \mathbb{R}^p \to \mathbb{R}$ that minimizes the expected error on data drawn from $D$:

$$\underset{h}{\text{minimize}} \underset{(\mathbf{x}_i, \mathbf{y}_i) \sim D}{\mathbf{E}} \phi_\ell(-h(\mathbf{x}_i) + \underline{y_i}) + \phi_\ell(h(\mathbf{x}_i) - \overline{y_i}),$$

Notice that, if $\ell(x) = x^2$, this is a generalization of the mean squared error to interval outputs. Moreover, this can be seen as a surrogate to a zero-one loss that measures if a predicted value lies within the target interval (Rigaill *et al.*, 2013).

### 3.2 Maximum margin interval trees

We will seek an interval *regression tree* model $T : \mathbb{R}^p \to \mathbb{R}$ that minimizes the total hinge loss on data set $S$:

$$C(T) \stackrel{\text{def}}{=} \sum_{(\mathbf{x}_i, \mathbf{y}_i) \in S} \left[ \phi_\ell \left( -T(\mathbf{x}_i) + \underline{y_i} + \epsilon \right) + \phi_\ell \left( T(\mathbf{x}_i) - \overline{y_i} + \epsilon \right) \right], \tag{1}$$

where $\epsilon \in \mathbb{R}_0^+$ is a hyperparameter introduced to improve regularity (see supplementary material for details).

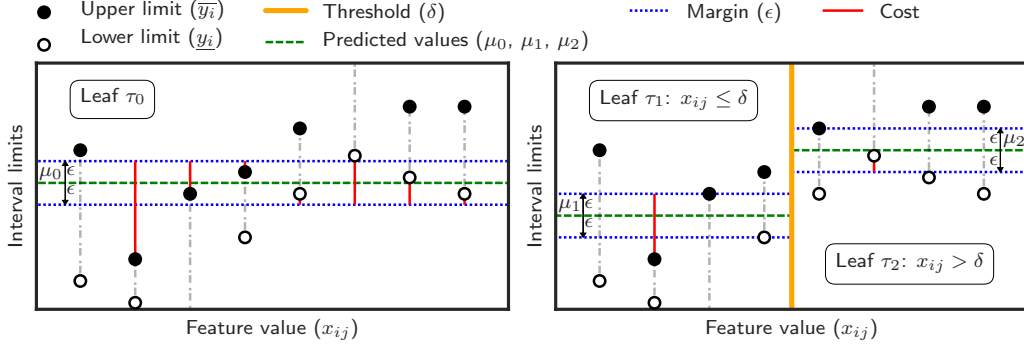

Figure 1: An example partition of leaf $\tau_0$ into leaves $\tau_1$ and $\tau_2$.

A *decision tree* is an arrangement of nodes and leaves. The leaves are responsible for making predictions, whereas the nodes guide the examples to the leaves based on the outcome of some boolean-valued rules (Breiman *et al.*, 1984). Let $\widetilde{T}$ denote the set of leaves in a decision tree $T$. Each leaf $\tau \in \widetilde{T}$ is associated with a set of examples $S_\tau \subseteq S$, for which it is responsible for making predictions. The sets $S_\tau$ obey the following properties: $S = \bigcup_{\tau \in \widetilde{T}} S_\tau$ and $S_\tau \cap S_{\tau'} \neq \emptyset \Leftrightarrow \tau = \tau'$. Hence, the contribution of a leaf $\tau$ to the total loss of the tree $C(T)$, given that it predicts $\mu \in \mathbb{R}$, is

$$C_\tau(\mu) \stackrel{\text{def}}{=} \sum_{(\mathbf{x}_i, \mathbf{y}_i) \in S_\tau} \left[ \phi_\ell(-\mu + \underline{y_i} + \epsilon) + \phi_\ell(\mu - \overline{y_i} + \epsilon) \right] \tag{2}$$

and the optimal predicted value for the leaf is obtained by minimizing this function over all $\mu \in \mathbb{R}$.

As in the CART algorithm (Breiman *et al.*, 1984), our tree growing algorithm relies on recursive partitioning of the leaves. That is, at any step of the tree growing algorithm, we obtain a new tree $T'$ from $T$ by selecting a leaf $\tau_0 \in \widetilde{T}$ and dividing it into two leaves $\tau_1, \tau_2 \in \widetilde{T'}$, s.t. $S_{\tau_0} = S_{\tau_1} \cup S_{\tau_2}$ and $\tau_0 \notin \widetilde{T'}$. This partitioning results from applying a boolean-valued rule $r : \mathbb{R}^p \to \mathbb{B}$ to each example $(\mathbf{x}_i, \mathbf{y}_i) \in S_{\tau_0}$ and sending it to $\tau_1$ if $r(\mathbf{x}_i) = \text{True}$ and to $\tau_2$ otherwise. The rules that we consider are threshold functions on the value of a single feature, i.e., $r(\mathbf{x}_i) \stackrel{\text{def}}{=}$ "$x_{ij} \leq \delta$". This is illustrated in Figure 1. According to Equation (2), for any such rule, we have that the total hinge loss for the examples that are sent to $\tau_1$ and $\tau_2$ are

$$C_{\tau_1}(\mu) = \overleftarrow{C_{\tau_0}}(\mu|j, \delta) \stackrel{\text{def}}{=} \sum_{(\mathbf{x}_i, \mathbf{y}_i) \in S_{\tau_0} : x_{ij} \leq \delta} \left[ \phi_\ell(-\mu + \underline{y_i} + \epsilon) + \phi_\ell(\mu - \overline{y_i} + \epsilon) \right] \tag{3}$$

$$C_{\tau_2}(\mu) = \overrightarrow{C_{\tau_0}}(\mu|j, \delta) \stackrel{\text{def}}{=} \sum_{(\mathbf{x}_i, \mathbf{y}_i) \in S_{\tau_0} : x_{ij} > \delta} \left[ \phi_\ell(-\mu + \underline{y_i} + \epsilon) + \phi_\ell(\mu - \overline{y_i} + \epsilon) \right] . \tag{4}$$

The best rule is the one that leads to the smallest total cost $C(T')$. This rule, as well as the optimal predicted values for $\tau_1$ and $\tau_2$, are obtained by solving the following optimization problem:

$$\operatorname*{argmin}_{j, \delta, \mu_1, \mu_2} \left[ \overleftarrow{C_{\tau_0}}(\mu_1|j, \delta) + \overrightarrow{C_{\tau_0}}(\mu_2|j, \delta) \right] . \tag{5}$$

In the next section we propose a dynamic programming algorithm for this task.

## 4 Algorithm

First note that, for a given $j, \delta$, the optimization separates into two convex minimization sub-problems, which each amount to minimizing a sum of convex loss functions:

$$\min_{j, \delta, \mu_1, \mu_2} \left[ \overleftarrow{C_\tau}(\mu_1|j, \delta) + \overrightarrow{C_\tau}(\mu_2|j, \delta) \right] = \min_{j, \delta} \left[ \min_{\mu_1} \overleftarrow{C_\tau}(\mu_1|j, \delta) + \min_{\mu_2} \overrightarrow{C_\tau}(\mu_2|j, \delta) \right] . \tag{6}$$

We will show that if there exists an efficient dynamic program $\Omega$ which, given any set of hinge loss functions defined over $\mu$, computes their sum and returns the minimum value, along with a minimizing value of $\mu$, the minimization problem of Equation (6) can be solved efficiently.

Observe that, although there is a continuum of possible values for $\delta$, we can limit the search to the values of feature $j$ that are observed in the data (i.e., $\delta \in \{x_{ij} \, ; \, i = 1, \dots, n\}$), since all other values do not lead to different configurations of $S_{\tau_1}$ and $S_{\tau_2}$. Thus, there are at most $n_j \leq n$ unique thresholds to consider for each feature. Let these thresholds be $\delta_{j,1} < \dots < \delta_{j,n_j}$. Now, consider $\Phi_{j,k}$ as the set that contains all the losses $\phi_\ell(-\mu + \underline{y_i} + \epsilon)$ and $\phi_\ell(\mu - \overline{y_i} + \epsilon)$ for which we have $(\mathbf{x}_i, \mathbf{y}_i) \in S_{\tau_0}$ and $x_{ij} = \delta_{j,k}$. Since we now only consider a finite number of $\delta$-values, it follows from Equation (3), that one can obtain $\overleftarrow{C_\tau}(\mu_1|j, \delta_{j,k})$ from $\overleftarrow{C_\tau}(\mu_1|j, \delta_{j,k-1})$ by adding all the losses in $\Phi_{j,k}$. Similarly, one can also obtain $\overrightarrow{C_\tau}(\mu_1|j, \delta_{j,k})$ from $\overrightarrow{C_\tau}(\mu_1|j, \delta_{j,k-1})$ by removing all the losses in $\Phi_{j,k}$ (see Equation (4)). This, in turn, implies that $\min_\mu \overleftarrow{C_\tau}(\mu|j, \delta_{j,k}) = \Omega(\Phi_{j,1} \cup \dots \cup \Phi_{j,k})$ and $\min_\mu \overrightarrow{C_\tau}(\mu|j, \delta_{j,k}) = \Omega(\Phi_{j,k+1} \cup \dots \cup \Phi_{j,n_j})$.

Hence, the cost associated with a split on each threshold $\delta_{j,k}$ is given by:

$$
\begin{array}{rlcl}
\delta_{j,1}: & \Omega(\Phi_{j,1}) & + & \Omega(\Phi_{j,2} \cup \cdots \cup \Phi_{j,n_j}) \\
\dots & \dots & & \dots \\
\delta_{j,i}: & \Omega(\Phi_{j,1} \cup \cdots \cup \Phi_{j,i}) & + & \Omega(\Phi_{j,i+1} \cup \cdots \cup \Phi_{j,n_j}) \\
\dots & \dots & & \dots \\
\delta_{j,n_j-1}: & \Omega(\Phi_{j,1} \cup \cdots \cup \Phi_{j,n_j-1}) & + & \Omega(\Phi_{j,n_j})
\end{array}
\tag{7}
$$

and the best threshold is the one with the smallest cost. Note that, in contrast with the other thresholds, $\delta_{j,n_j}$ needs not be considered, since it leads to an empty leaf. Note also that, since $\Omega$ is a dynamic program, one can efficiently compute Equation (7) by using $\Omega$ twice, from the top down for the first column and from the bottom up for the second. Below, we propose such an algorithm.

## 4.1 Definitions

A general expression for the hinge losses $\phi_\ell(-\mu + \underline{y_i} + \epsilon)$ and $\phi_\ell(\mu - \overline{y_i} + \epsilon)$ is $\phi_\ell(s_i(\mu - y_i) + \epsilon)$, where $s_i = -1$ or $1$ respectively. Now, choose any convex function $\ell : \mathbb{R} \to \mathbb{R}$ and let

$$
P_t(\mu) \overset{\text{def}}{=} \sum_{i=1}^{t} \phi_\ell(s_i(\mu - y_i) + \epsilon)
\tag{8}
$$

be a sum of $t$ hinge loss functions. In this notation, $\Omega(\Phi_{j,1} \cup \dots \cup \Phi_{j,i}) = \min_\mu P_t(\mu)$, where $t = |\Phi_{j,1} \cup \dots \cup \Phi_{j,i}|$.

**Observation 1.** Each of the $t$ hinge loss functions has a breakpoint at $y_i - s_i\epsilon$, where it transitions from a zero function to a non-zero one if $s_i = 1$ and the converse if $s_i = -1$.

For the sake of simplicity, we will now consider the case where these breakpoints are all different; the generalization is straightforward, but would needlessly complexify the presentation (see the supplementary material for details). Now, note that $P_t(\mu)$ is a convex piecewise function that can be uniquely represented as:

$$
P_t(\mu) = \begin{cases}
p_{t,1}(\mu) & \text{if } \mu \in (-\infty, b_{t,1}] \\
\dots \\
p_{t,i}(\mu) & \text{if } \mu \in (b_{t,i-1}, b_{t,i}] \\
\dots \\
p_{t,t+1}(\mu) & \text{if } \mu \in (b_{t,t}, \infty)
\end{cases}
\tag{9}
$$

where we will call $p_{t,i}$ the $i^{\text{th}}$ piece of $P_t$ and $b_{t,i}$ the $i^{\text{th}}$ breakpoint of $P_t$ (see Figure 2 for an example). Observe that each piece $p_{t,i}$ is the sum of all the functions that are non-zero on the interval $(b_{t,i-1}, b_{t,i}]$. It therefore follows from Observation 1 that

$$
p_{t,i}(\mu) = \sum_{j=1}^{t} \ell[s_j(\mu - y_j) + \epsilon] \, I[(s_j = -1 \wedge b_{t,i-1} < y_j + \epsilon) \vee (s_j = 1 \wedge y_j - \epsilon < b_{t,i})]
\tag{10}
$$

where $I[\cdot]$ is the (Boolean) indicator function, i.e., $I[\text{True}] = 1$ and $0$ otherwise.

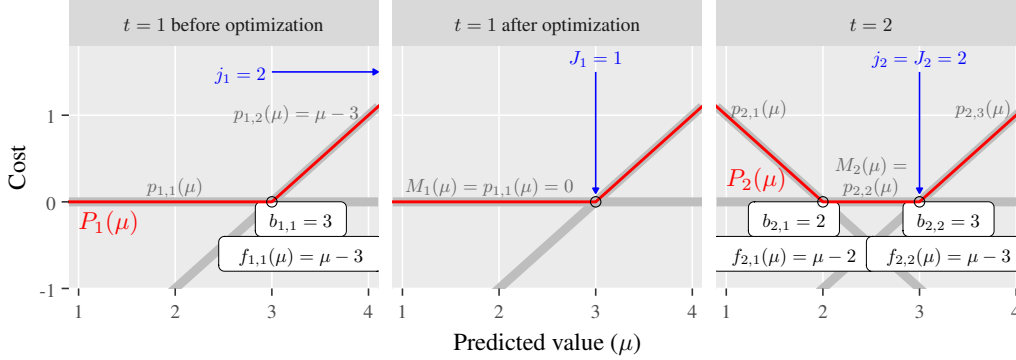

Figure 2: First two steps of the dynamic programming algorithm for the data $y_1 = 4, s_1 = 1, y_2 = 1, s_2 = -1$ and margin $\epsilon = 1$, using the linear hinge loss ($\ell(x) = x$). **Left:** The algorithm begins by creating a first breakpoint at $b_{1,1} = y_1 - \epsilon = 3$, with corresponding function $f_{1,1}(\mu) = \mu - 3$. At this time, we have $j_1 = 2$ and thus $b_{1,j_1} = \infty$. Note that the cost $p_{1,1}$ before the first breakpoint is not yet stored by the algorithm. **Middle:** The optimization step is to move the pointer to the minimum ($J_1 = j_1 - 1$) and update the cost function, $M_1(\mu) = p_{1,2}(\mu) - f_{1,1}(\mu)$. **Right:** The algorithm adds the second breakpoint at $b_{2,1} = y_2 + \epsilon = 2$ with $f_{2,1}(\mu) = \mu - 2$. The cost at the pointer is not affected by the new data point, so the pointer does not move.

**Lemma 1.** *For any $i \in \{1, ..., t\}$, we have that $p_{t,i+1}(\mu) = p_{t,i}(\mu) + f_{t,i}(\mu)$, where $f_{t,i}(\mu) = s_k \ell[s_k(\mu - y_k) + \epsilon]$ for some $k \in \{1, ..., t\}$ such that $y_k - s_k\epsilon = b_{t,i}$.*

*Proof.* The proof relies on Equation (10) and is detailed in the supplementary material. □

## 4.2 Minimizing a sum of hinge losses by dynamic programming

Our algorithm works by recursively adding a hinge loss to the total function $P_t(\mu)$, each time, keeping track of the minima. To achieve this, we use a pointer $J_t$, which points to rightmost piece of $P_t(\mu)$ that contains a minimum. Since $P_t(\mu)$ is a convex function of $\mu$, we know that this minimum is global. In the algorithm, we refer to the segment $p_{t,J_t}$ as $M_t$ and the essence of the dynamic programming update is moving $J_t$ to its correct position after a new hinge loss is added to the sum.

At any time step $t$, let $B_t = \{(b_{t,1}, f_{t,1}), ..., (b_{t,t}, f_{t,t}) \mid b_{t,1} < ... < b_{t,t}\}$ be the current set of breakpoints ($b_{t,i}$) together with their corresponding difference functions ($f_{t,i}$). Moreover, assume the convention $b_{t,0} = -\infty$ and $b_{t,t+1} = \infty$, which are defined, but not stored in $B_t$.

The initialization ($t = 0$) is

$$B_0 = \{\}, \ J_0 = 1, \ M_0(\mu) = 0. \tag{11}$$

Now, at any time step $t > 0$, start by inserting the new breakpoint and difference function. Hence,

$$B_t = B_{t-1} \cup \{(y_t - s_t\epsilon, \ s_t \ell[s_t(\mu - y_t) + \epsilon])\}. \tag{12}$$

Recall that, by definition, the set $B_t$ remains sorted after the insertion. Let $j_t \in \{1, \ldots, t+1\}$, be the updated value for the previous minimum pointer ($J_{t-1}$) after adding the $t^{\text{th}}$ hinge loss (i.e., the index of $b_{t-1,J_{t-1}}$ in the sorted set of breakpoints at time $t$). It is obtained by adding 1 if the new breakpoint is before $J_{t-1}$ and 0 otherwise. In other words,

$$j_t = J_{t-1} + I[y_t - s_t\epsilon < b_{t-1,J_{t-1}}]. \tag{13}$$

If there is no minimum of $P_t(\mu)$ in piece $p_{t,j_t}$, we must move the pointer from $j_t$ to its final position $J_t \in \{1, ..., t+1\}$, where $J_t$ is the index of the rightmost function piece that contains a minimum:

$$J_t = \max_{i \in \{1, ..., t+1\}} i, \text{ s.t. } (b_{t,i-1}, b_{t,i}] \cap \{x \in \mathbb{R} \mid P_t(x) = \min_\mu P_t(\mu)\} \neq \emptyset. \tag{14}$$

See Figure 2 for an example. The minimum after optimization is in piece $M_t$, which is obtained by adding or subtracting a series of difference functions $f_{t,i}$. Hence, applying Lemma 1 multiple times,

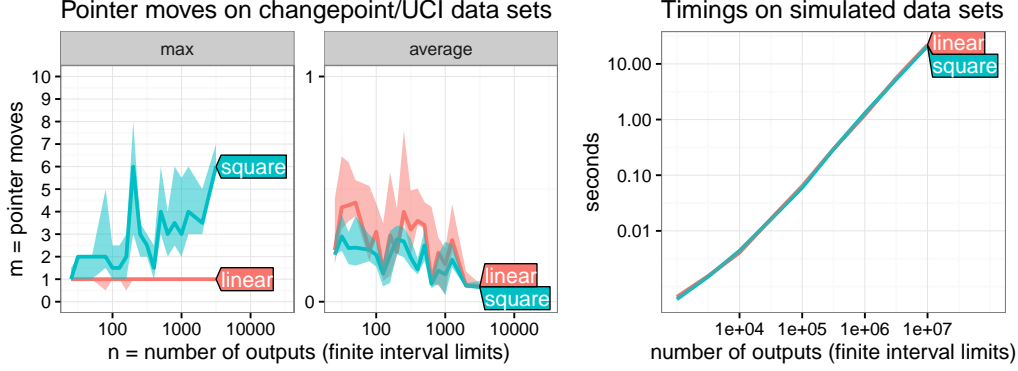

Figure 3: Empirical evaluation of the expected $O(n(m + \log n))$ time complexity for $n$ data points and $m$ pointer moves per data point. **Left:** max and average number of pointer moves $m$ over all real and simulated data sets we considered (median line and shaded quartiles over all features, margin parameters, and data sets of a given size). We also observed $m = O(1)$ pointer moves on average for both the linear and squared hinge loss. **Right:** timings in seconds are consistent with the expected $O(n \log n)$ time complexity.

we obtain:

$$
M_t(\mu) \stackrel{\text{def}}{=} p_{t,J_t}(\mu) = p_{t,j_t}(\mu) + \begin{cases} 0 & \text{if } j_t = J_t \\ \sum_{i=j_t}^{J_t-1} f_{t,i}(\mu) & \text{if } j_t < J_t \\ -\sum_{i=J_t}^{j_t-1} f_{t,i}(\mu) & \text{if } J_t < j_t \end{cases} \tag{15}
$$

Then, the optimization problem can be solved using $\min_\mu P_t(\mu) = \min_{\mu \in (b_{t,J_t-1}, b_{t,J_t}]} M_t(\mu)$. The proof of this statement is available in the supplementary material, along with a detailed pseudocode and implementation details.

### 4.3 Complexity analysis

The $\ell$ functions that we consider are $\ell(x) = x$ and $\ell(x) = x^2$. Notice that any such function can be encoded by three coefficients $a, b, c \in \mathbb{R}$. Therefore, summing two functions amounts to summing their respective coefficients and takes time $O(1)$. The set of breakpoints $B_t$ can be stored using any data structure that allows sorted insertions in logarithmic time (e.g., a binary search tree).

Assume that we have $n$ hinge losses. Inserting a new breakpoint at Equation (12) takes $O(\log n)$ time. Updating the $j_t$ pointer at Equation (13) takes $O(1)$. In contrast, the complexity of finding the new pointer position $J_t$ and updating $M_t$ at Equations (14) and (15) varies depending on the nature of $\ell$. For the case where $\ell(x) = x$, we are guaranteed that $J_t$ is at distance at most one of $j_t$. This is demonstrated in Theorem 2 of the supplementary material. Since we can sum two functions in $O(1)$ time, we have that the worst case time complexity of the linear hinge loss algorithm is $O(n \log n)$. However, for the case where $\ell(x) = x^2$, the worst case could involve going through the $n$ breakpoints. Hence, the worst case time complexity of the squared hinge loss algorithm is $O(n^2)$. Nevertheless, in Section 5.1, we show that, when tested on a variety real-world data sets, the algorithm achieved a time complexity of $O(n \log n)$ in this case also.

Finally, the space complexity of this algorithm is $O(n)$, since a list of $n$ breakpoints $(b_{t,i})$ and difference functions $(f_{t,i})$ must be stored, along with the coefficients $(a, b, c \in \mathbb{R})$ of $M_t$. Moreover, it follows from Lemma 1 that the function pieces $p_{t,i}$ need not be stored, since they can be recovered using the $b_{t,i}$ and $f_{t,i}$.

## 5 Results

### 5.1 Empirical evaluation of time complexity

We performed two experiments to evaluate the expected $O(n(m + \log n))$ time complexity for $n$ interval limits and $m$ pointer moves per limit. First, we ran our algorithm (MMIT) with both squared

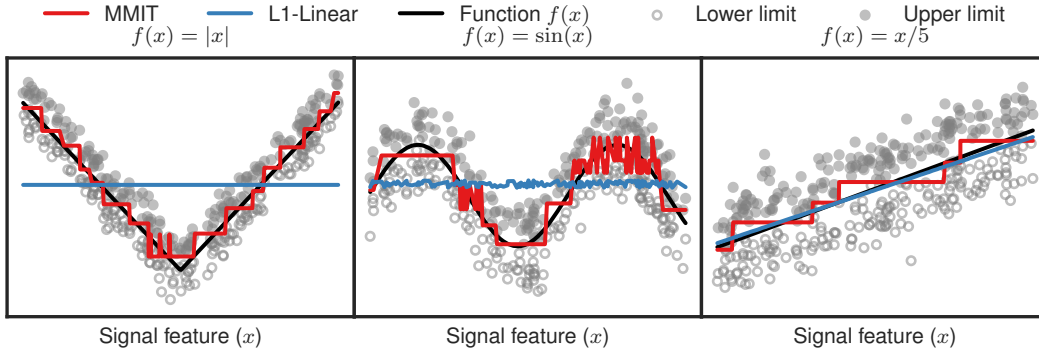

Figure 4: Predictions of MMIT (linear hinge loss) and the L1-regularized linear model of Rigaill *et al.* (2013) (L1-Linear) for simulated data sets.

and linear hinge loss solvers on a variety of real-world data sets of varying sizes (Rigaill *et al.*, 2013; Lichman, 2013), and recorded the number of pointer moves. We plot the average and max pointer moves over a wide range of margin parameters, and all possible feature orderings (Figure 3, left). In agreement with our theoretical result (supplementary material, Theorem 2), we observed a maximum of one move per interval limit for the linear hinge loss. On average we observed that the number of moves does not increase with data set size, even for the squared hinge loss. These results suggest that the number of pointer moves per limit is generally constant $m = O(1)$, so we expect an overall time complexity of $O(n \log n)$ in practice, even for the squared hinge loss. Second, we used the limits of the target intervals in the neuroblastoma changepoint data set (see Section 5.3) to simulate data sets from $n = 10^3$ to $n = 10^7$ limits. We recorded the time required to run the solvers (Figure 3, right), and observed timings which are consistent with the expected $O(n \log n)$ complexity.

## 5.2   MMIT recovers a good approximation in simulations with nonlinear patterns

We demonstrate one key limitation of the margin-based interval regression algorithm of Rigaill *et al.* (2013) (L1-Linear): it is limited to modeling linear patterns. To achieve this, we created three simulated data sets, each containing 200 examples and 20 features. Each data set was generated in such a way that the target intervals followed a specific pattern $f : \mathbb{R} \to \mathbb{R}$ according to a single feature, which we call the *signal feature*. The width of the intervals and a small random shift around the true value of $f$ were determined randomly. The details of the data generation protocol are available in the supplementary material. MMIT (linear hinge loss) and L1-Linear were trained on each data set, using cross-validation to choose the hyperparameter values. The resulting data sets and the predictions of each algorithm are illustrated in Figure 4. As expected, L1-Linear fails to fit the non-linear patterns, but achieves a near perfect fit for the linear pattern. In contrast, MMIT learns stepwise approximations of the true functions, which results from each leaf predicting a constant value. Notice the fluctuations in the models of both algorithms, which result from using irrelevant features.

## 5.3   Empirical evaluation of prediction accuracy

In this section, we compare the accuracy of predictions made by MMIT and other learning algorithms on real and simulated data sets.

**Evaluation protocol**   To evaluate the accuracy of the algorithms, we performed 5-fold cross-validation and computed the mean squared error (MSE) with respect to the intervals in each of the five testing sets (Figure 5). For a data set $S = \{(\mathbf{x}_i, \mathbf{y}_i)\}_{i=1}^n$ with $\mathbf{x}_i \in \mathbb{R}^p$ and $\mathbf{y}_i \in \overline{\mathbb{R}}^2$, and for a model $h : \mathbb{R}^p \to \mathbb{R}$, the MSE is given by

$$\text{MSE}(h, S) = \frac{1}{n} \sum_{i=1}^n \left( [h(\mathbf{x}_i) - \underline{y_i}] \, I[h(\mathbf{x}_i) < \underline{y_i}] + [h(\mathbf{x}_i) - \overline{y_i}] \, I[h(\mathbf{x}_i) > \overline{y_i}] \right)^2. \quad (16)$$

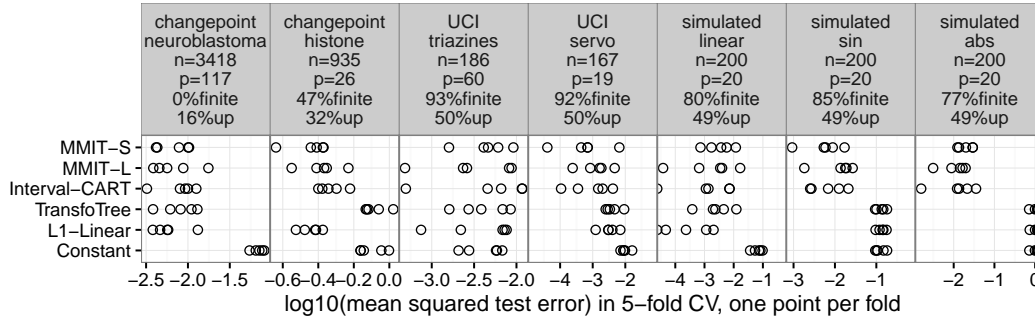

Figure 5: MMIT testing set mean squared error is comparable to, or better than, other interval regression algorithms in seven real and simulated data sets. Five-fold cross-validation was used to compute 5 test error values (points) for each model in each of the data sets (panel titles indicate data set source, name, number of observations=n, number of features=p, proportion of intervals with finite limits and proportion of all interval limits that are upper limits).

At each step of the cross-validation, another cross-validation (nested within the former) was used to select the hyperparameters of each algorithm based on the training data. The hyperparameters selected for MMIT are available in the supplementary material.

**Algorithms** The linear and squared hinge loss variants of Maximum Margin Interval Trees (MMIT-L and MMIT-S) were compared to two state-of-the-art interval regression algorithms: the margin-based $L1$-regularized linear model of Rigaill *et al.* (2013) (L1-Linear) and the Transformation Trees of Hothorn and Zeileis (2017) (TransfoTree). Moreover, two baseline methods were included in the comparison. To provide an upper bound for prediction error, we computed the trivial model that ignores all features and just learns a constant function $h(x) = \mu$ that minimizes the MSE on the training data (Constant). To demonstrate the importance of using a loss function designed for interval regression, we also considered the CART algorithm (Breiman *et al.*, 1984). Specifically, CART was used to fit a regular regression tree on a transformed training set, where each interval regression example $(\mathbf{x}, [\underline{y}, \overline{y}])$ was replaced by two real-valued regression examples with features $\mathbf{x}$ and labels $\underline{y} + \epsilon$ and $\overline{y} - \epsilon$. This algorithm, which we call Interval-CART, uses a margin hyperparameter and minimizes a squared loss with respect to the interval limits. However, in contrast with MMIT, it does not take the structure of the interval regression problem into account, i.e., it ignores the fact that no cost should be incurred for values predicted inside the target intervals.

**Results in changepoint data sets** The problem in the first two data sets is to learn a penalty function for changepoint detection in DNA copy number and ChIP-seq data (Hocking *et al.*, 2013; Rigaill *et al.*, 2013), two significant interval regression problems from the field of genomics. For the *neuroblastoma* data set, all methods, except the constant model, perform comparably. Interval-CART achieves the lowest error for one fold, but L1-Linear is the overall best performing method. For the *histone* data set, the margin-based models clearly outperform the non-margin-based models: Constant and TransfoTree. MMIT-S achieves the lowest error on one of the folds. Moreover, MMIT-S tends to outperform MMIT-L, suggesting that a squared loss is better suited for this task. Interestingly, MMIT-S outperforms Interval-CART, which also uses a squared loss, supporting the importance of using a loss function adapted to the interval regression problem.

**Results in UCI data sets** The next two data sets are regression problems taken from the UCI repository (Lichman, 2013). For the sake of our comparison, the real-valued outputs in these data sets were transformed into censored intervals, using a protocol that we detail in the supplementary material. For the difficult *triazines* data set, all methods struggle to surpass the Constant model. Nevethess, some achieve lower errors for one fold. For the *servo* data set, the margin-based tree models: MMIT-S, MMIT-L, and Interval-CART perform comparably and outperform the other models. This highlights the importance of developing non-linear models for interval regression and suggests a positive effect of the margin hyperparameter on accuracy.

**Results in simulated data sets** The last three data sets are the simulated data sets discussed in the previous section. As expected, the L1-linear model tends outperforms the others on the *linear* data set. However, surprisingly, on a few folds, the MMIT-L and Interval-CART models were able to achieve low test errors. For the non-linear data sets (*sin* and *abs*), MMIT-S, MMIT-L and Interval-Cart clearly outperform the TransfoTree, L1-linear and Constant models. Observe that the TransfoTree algorithm achieves results comparable to those of L1-linear which, in Section 5.2, has been shown to learn a roughly constant model in these situations. Hence, although these data sets are simulated, they highlight situations where this non-linear interval regression algorithm fails to yield accurate models, but where MMITs do not.

Results for more data sets are available in the supplementary material.

## 6 Discussion and conclusions

We proposed a new margin-based decision tree algorithm for the interval regression problem. We showed that it could be trained by solving a sequence of convex sub-problems, for which we proposed a new dynamic programming algorithm. We showed empirically that the latter's time complexity is log-linear in the number of intervals in the data set. Hence, like classical regression trees (Breiman *et al.*, 1984), our tree growing algorithm's time complexity is linear in the number of features and log-linear in the number of examples. Moreover, we studied the prediction accuracy in several real and simulated data sets, showing that our algorithm is competitive with other linear and nonlinear models for interval regression.

This initial work on Maximum Margin Interval Trees opens a variety of research directions, which we will explore in future work. We will investigate learning ensembles of MMITs, such as random forests. We also plan to extend the method to learning trees with non-constant leaves. This will increase the smoothness of the models, which, as observed in Figure 4, tend to have a stepwise nature. Moreover, we plan to study the average time complexity of the dynamic programming algorithm. Assuming a certain regularity in the data generating distribution, we should be able to bound the number of pointer moves and justify the time complexity that we observed empirically. In addition, we will study the conditions in which the proposed MMIT algorithm is expected to surpass methods that do not exploit the structure of the target intervals, such as the proposed Interval-CART method. Intuitively, one weakness of Interval-CART is that it does not properly model left and right-censored intervals, for which it favors predictions that are near the finite limits. Finally, we plan to extend the dynamic programming algorithm to data with un-censored outputs. This will make Maximum Margin Interval Trees applicable to survival analysis problems, where they should rank among the state of the art.

## Reproducibility

- Implementation: `https://git.io/mmit`
- Experimental code: `https://git.io/mmit-paper`
- Data: `https://git.io/mmit-data`

The versions of the software used in this work are also provided in the supplementary material.

## Acknowledgements

We are grateful to Ulysse Côté-Allard, Mathieu Blanchette, Pascal Germain, Sébastien Giguère, Gaël Letarte, Mario Marchand, and Pier-Luc Plante for their insightful comments and suggestions. This work was supported by the National Sciences and Engineering Research Council of Canada, through an Alexander Graham Bell Canada Graduate Scholarship Doctoral Award awarded to AD and a Discovery Grant awarded to FL (#262067).

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
