[Supplementary Material]

# Maximum Margin Interval Trees

## Supplementary material

Alexandre Drouin, Toby Dylan Hocking, François Laviolette

# Contents

Note to the reader: equations and figures that are unique to the supplementary material are prefixed with "S.", e.g., Equation (S.1). All other references refer to the main paper.

# 1 Section 3 - Problem

## 1.1 The effect of the margin hyperparameter

The margin ($\epsilon$) hyperparameter of Maximum Margin Interval Trees has a regularizing effect, which help prevents overfitting. This can be observed in Figure S.1, where the training and testing set mean squared errors are shown as a function of the margin. It can be observed that there exists a value of this hyperparameter for which the testing set error reaches a minimum, while the training set error slightly increases.

Intuitively, the margin makes the model more robust to noise by enforcing that its predictions must be at a certain distance of the training set interval limits, which are possibly noisy. This effect, combined with those of other regularization hyperparameters, such as the maximum depth of the decision tree and the minimum number of examples required for a leaf to be partitioned, make for the full regularization of our algorithm. The complex trade-off between these hyperparameters will be further studied in future work.

Figure S.1: Training and testing set mean squared errors for various margin sizes on the H3K36me3_AM_immune_FPOP changepoint detection data set.

# 2 Section 4 - Algorithm

## 2.1 Proof: Lemma 1

**Lemma 1.** *For any $i \in \{1,...,t\}$, we have that $p_{t,i+1}(\mu) = p_{t,i}(\mu) + f_{t,i}(\mu)$, where $f_{t,i}(\mu) = s_k \, \ell[s_k(\mu - y_k) + \epsilon]$ for some $k \in \{1,...,t\}$ such that $y_k - s_k\epsilon = b_{t,i}$.*

*Proof.* As described in Equation (9), the function pieces $p_{t,i}$ and $p_{t,i+1}$ are separated by the breakpoint $b_{t,i}$. Since we assume that each of the hinge losses has a unique breakpoint $y_j - s_j\epsilon$, $\forall j \in \{1,...,t\}$, we have that $b_{t,i}$ corresponds to the transition between a zero and non-zero state (or the converse) for a single hinge loss in the sum $P_t$. Denote this function $\phi_\ell(s_k(\mu - y_k) + \epsilon)$, where $k \in \{1,...,t\}$. We have that $b_{t,i} = y_k - s_k\epsilon$.

Moreover, it follows from Equation (10) that:

$$p_{t,i}(\mu) = \sum_{j=1}^{t} \ell[s_j(\mu - y_j) + \epsilon] \, I[(s_j = -1 \wedge b_{t,i-1} < y_j + \epsilon) \vee (s_j = 1 \wedge y_j - \epsilon < b_{t,i})] \quad \text{(S.1)}$$

and

$$p_{t,i+1}(\mu) = \sum_{j=1}^{t} \ell[s_j(\mu - y_j) + \epsilon] \, I[(s_j = -1 \wedge b_{t,i} < y_j + \epsilon) \vee (s_j = 1 \wedge y_j - \epsilon < b_{t,i+1})]. \quad \text{(S.2)}$$

There are two cases to consider: $s_k = -1$ and $s_k = 1$. If $s_k = -1$, we have that $p_{t,i+1}(\mu) - p_{t,i}(\mu) = -\ell[s_k(\mu - y_k) + \epsilon] = s_k \, \ell[s_k(\mu - y_k) + \epsilon]$, since $I[b_{t,i-1} < y_k + \epsilon = b_{t,i}] = 1$, but $I[b_{t,i} < y_k + \epsilon = b_{t,i}] = 0$. If $s_k = 1$, we have that $p_{t,i+1}(\mu) - p_{t,i}(\mu) = \ell[s_k(\mu - y_k) + \epsilon] = s_k \, \ell[s_k(\mu - y_k) + \epsilon]$, since $I[y_k - \epsilon < b_{t,i} = y_k - \epsilon] = 0$, but $I[y_k - \epsilon = b_{t,i} < b_{t,i+1}] = 1$.

Hence, we have shown that $p_{t,i+1}(\mu) - p_{t,i}(\mu) = s_k \, \ell[s_k(\mu - y_k) + \epsilon]$ and thus $p_{t,i+1}(\mu) = p_{t,i}(\mu) + s_k \, \ell[s_k(\mu - y_k) + \epsilon]$ for $k \in \{1,...,t\}$ such that $y_k - s_k\epsilon = b_{t,i}$.

$\square$

## 2.2  Proof: Optimality of the dynamic programming algorithm

**Theorem 1.** *At any time step (t) of the dynamic programming algorithm, we have that:*

$$\min_{\mu \in \mathbb{R}} P_t(\mu) = \min_{\mu \in (b_{t,J_t-1}, b_{t,J_t}]} M_t(\mu),$$

*where $P_t$ and the $b_{t,i}$ are defined at Equation (9), $J_t$ is defined at Equation (14), and $M_t$ is defined at Equation (15).*

*Proof.* Let $t$ be the current time step of the dynamic programming algorithm, i.e., the number of hinge losses in the sum $P_t(\mu)$. It follows from Equations (9) and (14) that the piece $p_{t,J_t}$ contains a global minimum of $P_t(\mu)$. Hence, we have that

$$\min_{\mu \in \mathbb{R}} P_t(\mu) = \min_{\mu \in (b_{t,J_t-1}, b_{t,J_t}]} P_t(\mu) = \min_{\mu \in (b_{t,J_t-1}, b_{t,J_t}]} p_{t,J_t}(\mu). \tag{S.3}$$

Moreover, it follows from a recursive application of Lemma 1 that, if $j_t < J_t$, we have

$$p_{t,j_t}(\mu) + \sum_{i=j_t}^{J_t-1} f_{t,i}(\mu) = p_{t,j_t+1}(\mu) + \sum_{i=j_t+1}^{J_t-1} f_{t,i}(\mu) = \cdots = p_{t,J_t}(\mu) \tag{S.4}$$

and, similarly, if $j_t > J_t$, we have $p_{t,J_t}(\mu) = p_{t,j_t}(\mu) - \sum_{i=J_t}^{j_t-1} f_{t,i}(\mu)$.

Hence, by Equation (15), we have that $M_t(\mu) = p_{t,J_t}(\mu)$ and thus, by Equation (S.3), we have that

$$\min_{\mu \in (b_{t,J_t-1}, b_{t,J_t}]} M_t(\mu) = \min_{\mu \in \mathbb{R}} P_t(\mu). \tag{S.5}$$

Therefore, the solution returned by the dynamic programming algorithm always corresponds to a global minimum of the sum of hinge losses $P_t$. $\quad\square$

## 2.3   Proof: Number of pointer moves for the linear hinge loss

**Theorem 2.** *Assuming that all hinge losses are linear, i.e., that $\ell(x) = x$, the insertion of a new hinge loss $\phi_\ell(s_t(\mu - y_t) + \epsilon)$ in the dynamic programming algorithm leads to a pointer $j_t$ that is at most at distance one of the optimal minimum pointer $J_t$ defined at Equation (14).*

*Proof.* First, observe that, according to Equation (10), we have that the gradient of any function piece $p_{t,i}$ with respect to $\mu \in \mathbb{R}$ is constant and given by:

$$\nabla p_{t,i}(\mu) = \sum_{j=1}^{t} s_j \, I[(s_j = -1 \wedge b_{t,i-1} < y_j + \epsilon) \vee (s_j = 1 \wedge y_j - \epsilon < b_{t,i})]. \tag{S.6}$$

This corresponds to the difference between the number of upper and lower interval limits for which the hinge loss takes a non-zero value on the segment $\mu \in (b_{t,i-1}, b_{t,i}]$. Hence, since $s_j \in \{-1, +1\}$, we have that the gradient of any function piece is a constant function whose value is an integer.

Moreover, it follows from Equation (S.6) and the fact that $b_{t,i} < b_{t,i+1}$ that:

$$\nabla p_{t,i}(\mu) < \nabla p_{t,i+1}(\mu), \; \forall \mu \in \mathbb{R}, i \in \{1, ..., t\}. \tag{S.7}$$

Indeed, going from $b_{t,i}$ to $b_{t,i+1}$, any function with sign $s_j = -1$ can only stop contributing negatively to the sum. Conversely, any function with sign $s_j = 1$ can only start contributing positively to the sum. Hence, we have that $\nabla p_{t,i}(\mu) \leq \nabla p_{t,i+1}(\mu)$, but since each $b_{t,i}$ corresponds to a change in the total function $P_t$, we have that $\nabla p_{t,i}(\mu) = \nabla p_{t,i+1}(\mu)$ is not possible and thus, $\nabla p_{t,i}(\mu) < \nabla p_{t,i+1}(\mu)$.

In addition, since $P_t$ is a piecewise linear function, it is not differentiable at its breakpoints $(b_{t,i})$. However, the subdifferential of $P_t$ at any $b_{t,i}$ is bounded by the gradient of the function piece on its left $(p_{t,i})$ and the one on its right $(p_{t,i+1})$, which are constant. Hence, we have the following subdifferential at any $b_{t,i}$:

$$\partial P_t(b_{t,i}) = [\nabla p_{t,i}(\mu), \nabla p_{t,i+1}(\mu)]. \tag{S.8}$$

Also, note that, if there exists a $\mu \in (b_{t,i-1}, b_{t,i}]$ such that $P_t(\mu) = \min_{\mu'} P_t(\mu')$, we have that:

$$0 \in \partial P_t(b_{t,i}) = [\nabla p_{t,i}(\mu), \nabla p_{t,i+1}(\mu)]. \tag{S.9}$$

Hence, by the definition of the minimum pointer $J_t$, given at Equation (14), we have that $0 \in \partial P_t(b_{t,J_t})$ and thus $\nabla p_{t,J_t}(\mu) \leq 0$. Moreover, we have that $\nabla p_{t,J_t+1}(\mu) \geq 1$, since $J_t$ points to the greatest breakpoint that has zero in its subdifferential and since the $\nabla p_{t,i}(\mu)$ are constant functions with integer values.

Now, we must consider two cases: the one where the inserted hinge loss has sign $s_t = -1$ and the one where it has sign $s_t = 1$.

- **Case $\mathbf{s_t = 1}$:** In this case, the pointer either remains unchanged or moves to the left. The latter situation only occurs when, after inserting the new hinge loss, we have $0 \notin \partial P_t(b_{t,j_t})$, which is only possible if $y_t - \epsilon < b_{t-1,J_{t-1}} = b_{t,j_t}$.

Observe that, according to Equation (S.6), the insertion of a new hinge loss with $s_t = 1$ can only increase the gradient of any segment by 1. Hence, if $y_t - \epsilon < b_{t,j_t}$, we have that $\nabla p_{t,j_t}(\mu) = \nabla p_{t-1,J_{t-1}}(\mu) + 1$.

If the insertion results in $\nabla p_{t,j_t}(\mu) \leq 0$, no pointer moves are required. However, if we have $\nabla p_{t,j_t}(\mu) = 1$, we have that $0 \notin \partial P_t(b_{t,j_t}) = [\nabla p_{t,j_t}(\mu), \nabla p_{t,j_t+1}(\mu)] = [1, \nabla p_{t,j_t+1}(\mu)]$. Thus, there are no values in $\mu \in (b_{t,j_t-1}, b_{t,j_t}]$ that are minimizers of $P_t(\mu)$ and the pointer must be moved left. It follows from Equation (S.7) that $\nabla p_{t,j_t-1}(\mu) < 1$ and thus $\nabla p_{t,j_t-1}(\mu) \leq 0$. Hence, we have that $0 \in \partial P_t(b_{t,j_t-1}) = [\nabla p_{t,j_t-1}(\mu), \nabla p_{t,j_t}(\mu)]$. Thus, by moving the pointer to the left once, i.e. $J_t = j_t - 1$, we have that $J_t$ is the largest value such that $p_{t,J_t}$ contains a minimum of $P_t(\mu)$ for $\mu \in (b_{t,J_t-1}, b_{t,J_t}]$, i.e., $0 \in \partial P_t(b_{t,J_t}) = [\nabla p_{t,J_t}(\mu), \nabla p_{t,J_t+1}(\mu)] = [\nabla p_{t,j_t-1}(\mu), \nabla p_{t,j_t}(\mu)]$.

- **Case $s_t = -1$:** In this case, the pointer either remains unchanged or moves to the right. The proof for this case is similar and left as an exercise to the reader.

$\square$

## 2.4 Pseudocode and Implementation details

We propose to store each linear/quadratic function $f(\mu) = a\mu^2 + b\mu + c$ in terms of its three coefficients $a, b, c \in \mathbb{R}$, where $a = 0$ for linear functions. Function sums can therefore be implemented in constant $O(1)$ time, by simply adding their coefficients (e.g., line 8 of Algorithm 1). Note that, in Algorithm 1, $B[J]$.breakpoint is the breakpoint at the pointer $J$, i.e., $b_{t,J}$ in the notation of Equation (9).

We propose to store the set of breakpoints $B_t$ using the map container of the C++ Standard Template Library ($B$ in Algorithm 1). It guarantees that the insertion of a breakpoint, described at Equation (12), takes $O(\log t)$ time (line 6). The pointers $j_t$ and $J_t$ can be implemented using a map::iterator ($J$ in Algorithm 1). Thus, the update rule given by Equation (13) happens automatically when the new breakpoint is inserted – the variable $J$ is $J_{t-1}$ before the insert, and it becomes $j_t$ after the insert.

The update rules for $J_t$ and $M_t$ (Equations (14) and (15)) are implemented in the while loop on lines 9–11 of Algorithm 1. Each iteration of the while loop is a constant $O(1)$ time operation. The MinInInterval sub-routine exploits the convexity of the cost function, and returns TRUE if a global minimum occurs on the function piece $M$, i.e. $p_{t,J}$ in the notation of Equation (9), whose limits are $B[J-1]$.breakpoint (i.e., $b_{t,J-1}$) and $B[J]$.breakpoint (i.e., $b_{t,J}$). If no minimum occurs on this piece, the pointer $J$ must be moved. If $M$ is increasing within the interval limits, the pointer should be moved left (line 10). Otherwise, it should be moved right (line 11).

Once the pointer has been moved to the interval that contains the minimum, the Minimize sub-routine returns an optimal prediction $\mu_t^*$ and cost $P_t^*$ (line 12) in constant $O(1)$ time.

An open-source implementation of Algorithm 1 is available at `https://git.io/mmit`.

---

**Algorithm 1** Dynamic programming algorithm for computing minimum total hinge loss.

---
1: Input: limits $\mathbf{y} \in \mathbb{R}^n$, signs $s \in \{-1, 1\}$, margin $\epsilon \in \mathbb{R}$.
2: Initialize: $B \leftarrow \text{map}\{\}$, $J \leftarrow B.\text{end}()$, $M \leftarrow \text{Coefs}(0)$
3: for data points $t$ from 1 to $n$:
4: $\quad f \leftarrow \text{Coefs}[s_t \ell(s_t(\mu - y_t) + \epsilon)]$
5: $\quad b \leftarrow y_t - s_t \epsilon$
6: $\quad B.\text{insert}(b, f)$
7: $\quad$ if $0 < s_t(B[J].\text{breakpoint} - y_t) + \epsilon$:
8: $\quad\quad M \leftarrow M + \text{Coefs}[\ell(s_t(\mu - y_t) + \epsilon)]$
9: $\quad$ while !MinInInterval$(M, B, J)$:
10: $\quad\quad$ if Increasing$(M)$: $J \leftarrow J - 1$; $M \leftarrow M - B[J].\text{function}$
11: $\quad\quad$ else: $M \leftarrow M + B[J].\text{function}$; $J \leftarrow J + 1$
12: $\quad \mu_t^*, P_t^* \leftarrow \text{Minimize}(M, B, J)$
13: Output: $\mu^* \in \mathbb{R}^n, P^* \in \mathbb{R}^n$

---

## 2.5 What if the breakpoints are not all different?

In the paper, we assumed that all the hinge losses in the sum of Equation (8) had distinct breakpoints (see Section 4.1). That is, we assumed that there were no interval limits with the exact same type (upper or lower limit) and value. This is not a very strong assumption, since the interval limits are real numbers (double precision on computers). However, it allows to simplify the presentation of the algorithm.

In fact, considering that some breakpoints could be equal leads to a total function $P_t$ (Equation (9)) with more pieces ($p_{t,i}$) than breakpoints ($b_{t,i}$). Consequently, this complicates the expressions of Equations (9 - 10) and Lemma 1. For example, in the statement of Lemma 1, the difference between $p_{t,i+1}$ and $p_{t,i}$ would no longer be a single $f_{t,i}$, but a sum of such values.

Nevertheless, note that the implementation provided with this work handles the case where some breakpoints are equal and the assumption is limited to the theoretical work.

# 3  Section 5 - Results

## 3.1  Protocol: Generating simulated data sets

Each of the simulated data set contains 200 learning examples, each represented by a vector of 20 features ($\mathbf{x}_i \in \mathbb{R}^{20}$) and a target interval ($\mathbf{y}_i = [\underline{y_i}, \overline{y_i}] \in \overline{\mathbb{R}}^2$). The feature vectors were generated by uniform random sampling in the range $[0, 10]$. Then, the target intervals were generated by applying some function $f : \mathbb{R} \to \mathbb{R}$ to the first feature of each example (i.e., $x_{i0}$ for the $i^{\text{th}}$ example), which we refer to as the *signal feature*. Specifically, ten values were sampled from a normal distribution $\mathcal{N}(f(x_{i0}), 0.3)$. The smallest value was used as the interval's lower limit ($\underline{y_i}$) and the maximum one as the interval's upper limit ($\overline{y_i}$). Then, a small vertical shift was added by sampling a value from $\mathcal{N}(0, 0.2)$ and adding it to both interval bounds. Finally, with probability 20%, one of the interval bounds was removed to simulate open interval. This therefore generates left, right and interval-censored data.

## 3.2  Protocol: Converting regression data sets to interval regression

Let $S \stackrel{\text{def}}{=} \{(\mathbf{x}_i, y_i)\}_{i=1}^n$, with $\mathbf{x}_i \in \mathbb{R}^p$ and $y_i \in \mathbb{R}$, be a real-valued regression data set. Moreover, let $S' \stackrel{\text{def}}{=} \{(\mathbf{x}_i, \mathbf{y}_i)\}_{i=1}^n$, with $\mathbf{x}_i \in \mathbb{R}^p$ and $\mathbf{y}_i = [\underline{y_i}, \overline{y_i}] \in \overline{\mathbb{R}}^2$, be its corresponding interval regression data set. The target intervals in $S'$ are generated by randomly sampling from normal distributions centered on the target values in $S$. Specifically, in this work, for any $y_i$, 10 values were sampled from a normal distribution $\mathcal{N}(y_i, \frac{y_i}{5})$. The smallest value was used as the interval's lower limit ($\underline{y_i}$) and the maximum one as the interval's upper limit ($\overline{y_i}$). Then, a small vertical shift was added by sampling a value from $\mathcal{N}(0, \frac{y_i}{10})$ and adding it to both interval bounds. Finally, with probability 10%, one of the interval bounds was removed to simulate open interval. The resulting data set $S'$ thus contains left, right and interval-censored data, which is derived from the true data in $S$.

## 3.3 Additional result: Benchmark results on more data sets

Figure S.2 shows a comparison of the learning algorithms, including our MMITs, on a wide variety of real and simulated data sets. The real data sets were taken from the work of Hocking *et al.* (2013), Rigaill *et al.* (2013), and from a repository maintained by the Connectionist Artificial Intelligence Laboratory (`https://github.com/renatopp/arff-datasets`). The latter repository includes many data sets from the UCI repository (Lichman, 2013). The simulated data sets are the three used in the paper. Note that the *neuroblastoma* changepoint detection data set used in the paper corresponds to *neuroblastomaProcessed* in Figure S.2, and that the *histone* one corresponds to *H3K27ac-H3K4me3_ TDHAM_BP_FPOP*.

*(See next page)*

Figure S.2: Mean-squared error (and standard deviation) for the 5 cross-validation folds. The data sets are sorted by decreasing number of examples.

## 3.4 Additional result: Selected hyperparameter values for MMIT

For each of the five cross-validation folds, the best hyperparameter values were chosen based on the training data, by performing 10-fold cross-validation over a grid of possible values.

Figure S.3: Margin values ($\epsilon$) for each of the five cross-validation folds.

Figure S.4: Minimum number of examples required to split a leaf, for each of the five cross-validation folds.

Figure S.5: Maximum tree depth for each of the five cross-validation folds.

## 3.5  Software versions

To ensure the reproducibility of our results, we list the versions of the software that was used to compare our algorithm to other methods:

1. Maximum margin interval trees:
   - mmit v1.1.1 (Python package)

2. Transformation trees:
   - trft v0.2-1 (R package)
   - partykit v1.2-0 (R package)

3. Interval-CART:
   - sklearn v0.18.1 (Python package)