[Reviews · NeurIPS 2017]

Reviewer 1



The paper introduces a computationally efficient decision tree algorithm for learning interval data outcomes. The paper is well written and provides useful illustrates to document the method. The method shows fine performance on simulated data and several real data sets. The paper compares against other interval based prediction methods. However, several of them are not contrasted in the related work. Adding differentiating factors from those would strengthen the paper in identifying when to use MMIT. While the paper is motivated heavily by survival analysis, the algorithm was not demonstrated on classic survival data sets (e.g. UCI thoracic surgery, PBC, etc.). It is not clear why not, given the motivation and that the algorithm appears to work with intervals defined as y_= y-. In survival analysis one is typically interested in rate estimation and risk attribution. In this case, the method would predict a time of event. I imagine risk attribution would be done in the same way as, e.g. random survival forests. Decision trees can be limited in their performance. Food for thought: how might you extend this to MMI forests?

Reviewer 2



The authors of this paper present a new decision tree algorithm for the interval regression problem. Leaves are partitioned using a margin based hinge loss similar to the L1-regularized hinge loss in Rigaill et al, Proc ICML 2013. However, the regression tree algorithm presented in this work is not limited to modeling linear patterns as the L1-regularized linear models in Rigaill et al. For training the non linear tree model, a sequence of convex optimization subproblems are optimally solved in log-linear time by Dynamic Programming (DP). The new maximum margin interval tree (MMIT) algorithm is compared with state-of-the-art margin-based and non-margin-based methods in several real and simulated datasets. - In terms of originality, the proposed margin based hinge loss is similar to the L1-regularized hinge loss in Rigaill et al, Proc ICML 2013. However, the regression tree algorithm presented in this work is not limited to modeling linear patterns as the L1-regularized linear models in Rigaill et al. MMIT achieves low test errors on nonlinear datasets and yields accurate models on simulated linear datasets as well. - In terms of significance: interval regression is a fundamental problem in fields such as survival analysis and computational biology. There are very few algorithms designed for this task, and most of them are linear models. A method learning nonlinear tree models like Maximum Margin Interval Trees (MMIT) could be helpful for practitioners in the field. - It terms of quality, the paper is fairly executed, the authors compare MMIT with state-of-the-art margin-based and non-margin-based methods in several real and simulated datasets. The 11-page supplementary material contains proofs, pseudocode, details of the open source implementation (link was hidden for anonymity) and of the experiments. - In terms of clarity, this is a well written paper. In summary, my opinion is that this is a well written and nicely organized work; the proposed method would be useful in real-world applications, and the novelty of the work satisfies the NIPS standards. Thus I recommend this work for publication.

Reviewer 3



Maximum Margin Interval Trees --------------------------------- In this paper the authors study interval regression problems, where each example is associated with a range output instead of a single point. Specifically, the authors investigate how to modify trees to produce such output by minimizing a modified sum of hinge losses. The key contribution of the paper is a dynamic programming algorithm that efficiently constructs these trees. The authors provide experimental results on a variety of simulated and real datasets. This is a generally well written paper, though it gets a little subscriptitis in 4.1-4.2. The algorithm seems sound and the experiments do a good job of comparing the approach to a reasonable set of baselines on a variety of datasets along both accuracy and time complexity metrics (which is the key selling point of the paper). I did have some questions that the authors could clarify: 1. What is epsilon in the experiments? How was it chosen? What is the effect of varying it? 2. What changes if the breakpoints are not all different (line 117)? 3. Does the algorithm work for trees with nonconstant leaves? (a model tree of some sort) What would need to change? 4. The choice of the examples with CART in the experiments seems rather strange. I think a baseline with CART where the examples are (x, (y_low+y_high)/2) would make more sense. 5. Along the same lines, would just building two trees one for y_low and the other for y_high work? I'm not very convinced we need a separate method just for this kind of problem. To summarize, this is a nice paper that proposes and studies an algorithm for learning interval trees and supports it with experimental results. Some key clarifications and experimental modifications would make the paper stronger. After feedback: The authors have clarified several questions I had, and I have upgraded my score to take this into account.